# Grenadian cats as potential reservoir for *Leptospira*

**Keith K. Kalasi, Daniel Fitzpatrick, Diana Stone, Talia Guttin, Andy Alhassan** [ORCID] *

St. George's University, School of Veterinary Medicine, Department of Pathobiology, Grenada, West Indies

* aalhassa@sgu.edu

**Data Availability Statement:** All relevant data are within the paper and its Supporting Information files.

**Funding:** This research was funded through a small grant awarded by WINDREF at St. Georges

## Abstract

Leptospirosis is a spirochetal disease caused by *Leptospira* spp. bacteria with global distribution affecting multiple mammalian species, including humans. The disease is endemic in many geographic areas and is of particular concern in tropical regions with abundant rainfall, such as the Caribbean. Currently, there are no published reports on *Leptospira* exposure or infection in Grenadian cat populations, and the role of cats in the epidemiology of the disease is unknown. Our hypothesis is that Grenadian cat population may be a reservoir host for zoonotic *Leptospira* on the island. During 2019–2021, 160 feral cats were captured from three parishes in Grenada, and their urine and blood were tested for *Leptospira*. Urine from 4 of 148 (2.7%) cat samples tested PCR-positive for *Leptospira*. Serum from 6 of 136 (4.4%) cat samples tested PCR-positive for *Leptospira*. Sequence analysis of the *Leptospira rpoB* gene showed 86–100% similarity to known or presumed pathogenic *Leptospira* species. The findings of this study suggest that Grenadian cat populations are infected with and are shedding *Leptospira* genotypes that are phylogenetically related to known pathogenic *Leptospira*, including known zoonotic strains. These observations support the hypothesis that Grenadian cat populations may be a reservoir host for zoonotic *Leptospira* on the Island.

## Author summary

Leptospirosis caused by pathogenic strains of *Leptospira* spp. is one of the most widespread zoonoses of humans and infections in animals. Rodents are considered the main reservoir hosts of *Leptospira*. The role of cats in the epidemiology of zoonotic *Leptospira* is not well-understood. Several recent studies confirm that cats can be infected by *Leptospira*, seroconvert, and shed spirochetes in their urine. Renal carriage and urine shedding can persist in cats even after clearance of the organism from the blood. There are few reports of human leptospirosis in Grenada. However, there are no published reports on *Leptospira* exposure or infection in Grenadian cat populations, and the role of cats in the epidemiology of the disease is unknown. We hypothesize that Grenadian cat population may be a reservoir host for zoonotic *Leptospira* on the island. The results show that feral cats are likely to be important reservoirs of pathogenic *Leptospira* spp. and a source of leptospirosis.

University, Grenada. The funders had no role in the study design, data collection and analysis, decision to publish, or preparation of the manuscript.

**Competing interests:** The authors have declared that no competing interests exist.

## Introduction

Leptospirosis is caused by a spirochete bacterium of the genus *Leptospira* [1,2,3,4], with more than a million reported human cases worldwide annually [1,3,5,6]. It is endemic in many geographic areas but is of major concern in tropical and subtropical regions such as the Caribbean, where there is significant rainfall and an abundance of the main reservoir hosts, rodents [5,6,7,8]. Rodents in at least nine genera are historically considered the main reservoir hosts of *Leptospira* and are often asymptomatic throughout the course of infection [9]. However, infection in a range of additional hosts including Carnivora, Didelphimorphia, Cetacea, Cingulata, Afrosoricida, Chiroptera and Primate orders, as well as in Reptilia and Amphibia classes has been documented [9].

*Leptospira* is classified into 64 species which are grouped into two major clades, pathogenic (P) and saprophytic (S), and each clade is subdivided into 2 subclades (P1, P2. S1 and S2) [10]. The clade P1 originally represented the pathogenic lineage, which contains virulent strains able to cause severe disease in humans and animals [10,11,12,13]; P2 is the intermediates consisting of species of low virulence causing mild to moderate, self-limiting illness without serious disease or death [10,14,15]; and S1 is the original saprophytic lineage of free living microorganisms that are not known to cause disease [10,12,13]; S2 is the new saprophytic clade.

The pathophysiology and lesions associated with leptosirosis are well-characterized in many mammals, including humans [2,7,16]. However, the pathogenesis of leptospiral disease is not completely understood [1,7,17]. Depending on the host and infecting serovar, leptospiral infection may cause a spectrum of syndromes from asymptomatic carriers to fulminant, acute disease [18]. Known reservoir hosts, like rodents, tend to be asymptomatic, harbor spirochetes in their renal tubules, and shed organisms into the environment via urine [1,2,4,6,7,13,19,20,21,22]. Transmission of *Leptospira* most commonly occurs by contact with contaminated water and soil but can also occur by direct contact with blood and urine from infected animals [6,11,16,19,20,21,23,24,25,26,27]. On average, the incubation period for leptospirosis is 5–14 days but can be as long as 30 days [7,11,17,28]. Infection with pathogenic *Leptospira* spp. can also lead to persistent renal carriage in reservoir hosts leading to shedding of leptospires in high concentrations [11,20,29,30,31]. Continuous shedding of leptospires into the environment by reservoir hosts and the bacteria's ability to survive for weeks to several months in soil and water makes *Leptospira* a significant public health problem, especially in tropical regions [1,2,11,18,32,33,34,35].

In humans, leptospirosis causes a wide variety of clinical manifestations, ranging from mild flu-like illness to severe and sometimes fatal disease [7,16,33]. About 90% of recognized human cases are mild and self-limited, but there are several severe manifestations [7]. The hallmark lesion of severe leptospirosis in non-reservoir hosts is vascular endothelial damage [7,11,22,33,36]. A severe form of the disease known as Weil's disease is characterized by jaundice, acute renal failure, aseptic meningitis, and hemorrhagic diathesis [11,17,27,37]. Globally, leptospirosis is thought to be even more prevalent than other zoonotic diseases; however, it is frequently underdiagnosed and underreported due to its nonspecific symptoms, which mimic better-known diseases [5,38,39]. Additionally, inadequate active surveillance and lack of appropriate diagnostic tests, such as serology and molecular tests, further contribute to the under-reporting of the disease [5,6,7,33].

### *Leptospira* infection in cats

The role of cats in the epidemiology of zoonotic *Leptospira* is not well-understood and has not received much attention [2,22,40,41,42]. In contrast to dogs and other mammalian species,

only a few reports of clinical cases of leptospirosis in cats have been published [2,20,31,43]. In dogs and other mammalian species, clinical signs range from mild symptoms such as lethargy, vomiting, anorexia, and polydipsia [44], to severe illness and death, often as a result of renal injury [2,36,43,45,46]. Clinical leptospirosis is rarely diagnosed in cats [2,18,20,31]. Furthermore, the clinical signs tend to be non-specific, such as ascites [47] and renal insufficiency [48]. Several recent studies confirm that cats can be infected by *Leptospira*, seroconvert, and shed spirochetes in their urine [2,18,20,21,22,26,31,32,40,41,42,43,48,49,50,51,52]. Renal carriage and urine shedding can persist in cats even after clearance of the organism from the blood and after antibody titers have dropped below the level of detection [20,40,50,53]. Hence cats serve as a source of infection and a sentinel for *Leptospira* [20].

Several studies around the world suggest that cats might be possible reservoir hosts for *Leptospira* and have reported PCR detection of *Leptospira* DNA in urine with the following results: 0.8% (2/260) PCR-positive of healthy indoor/outdoor cats in Thailand [54] 1.6% (2/125) PCR-positive of healthy cats and 5.3% (6/113) PCR-positive of cats with kidney disease in Canada [41]; 3.3% (7/215) PCR-positive of outdoor cats in Germany [40]; 3.5% (7/200) PCR-positive urine samples from feral cats in Prince Edwards Island, Canada [3]; 11.8% (10/85) PCR-positive urine samples from shelter cats in Colorado, USA [20]; 16.7% (4/24) PCR-positive urine samples from two shelters in Malaysia [55]. A high PCR prevalence of *Leptospira* in urine of 67.8% (80/118) was detected in cats in Taiwan which according to the authors of the study occurs during the typhoon and flood season [52]. In addition, the study showed that stray cats had a higher PCR-positive urine prevalence (77.2%) than owned cats (34.6%) [52], which might be due to the fact that stray cats come into contact with the *Leptospira* rodent reservoir hosts more flequesntly frequently than owned cats, perhaps through hunting behavior [21,42,43,49,50].

## Leptospirosis in Grenada

Grenada is a tropical island in the southern part of the West Indies with ideal conditions for environmental survival of *Leptospira* (e.g., year-round warm weather and rain as well as an abundance of known or suspected reservoir hosts). Additionally, similar to many other Caribbean Islands, Grenada has the small Indian mongoose (*Herpestes auropunctatus*), an invasive animal species which has a potential for maintenance or amplification of zoonotic pathogens such as *Leptospira*, Rabies virus, *Salmonella*, *and Campylobacter* [8,56,57]. Between 2008 and 2014, Grenada reported 2–22 cases of human leptospirosis annually, approximate population of 108,000 [58]. This likely represents a small fraction of cases due to low clinical suspicion, difficulty in distinguishing clinical leptospirosis from other endemic diseases, and a shortage of diagnostic laboratory capacity [6,59,60]. Importantly, there have not been any attempts to date to isolate *Leptospira* from human cases in Grenada, and thus, the pathogenic strains infecting humans on the island are unknown. Humans and animals in Grenada have tested positive for antibodies against at least 17 serotypes of *Leptospira*, likely reflecting the major zoonotic species (*L. borgpetersenii*, *L. interrogans*, *L. kirschneri*, and *L. noguchii*).

Currently, there are no published reports on feline leptospirosis or exposure of cats to *Leptospira* in Grenada. Grenada's feral cats are common in and around households, and they are more likely than owned cats to have exposure to the main reservoir host (rodents). Due to their proximity to humans and domestic animals, it is important to determine if Grenada's feral cats are exposed to *Leptospira* and whether they are shedding *Leptospira* in their urine, which, if present, could be contaminating the environment and contributing to *Leptospira* transmission. The aim of this study is to determine if Grenadian cats are exposed to, infected with, and shedding pathogenic and potentially zoonotic *Leptospira* in their urine.

## Materials and methods

### Ethics statement

The study was conducted according to the guidelines approved by the Institutional Animal Care and Use committee of St. George's University (IACUC #18016-R dated November 9[th], 2018).

### a) Feral cat trapping

Between January 2019 and August 2021, 160 healthy male and female feral (community cats) cats were captured with Havahart live traps from three parishes in the island of Grenada by the Saint Georges University, School of Veterinary Medicine (SGU SVM) Feral Cat Project (FCP) group.

The minimum sample size required (75 cats) to detect a 12.0% total prevalence (exposed, infected, and shedding) or greater for *Leptospira* exposure/infection/shedding was determined using the Population Proportion calculator with 5.0% margin of error and 95.0% confidence interval [61].

The FCP utilizes an established protocol approved by the SGU Institutional Animal Care and Use Committee (IACUC) for trapping, handling, and treating feral cats (SGU IACUC-15022-T and IACUC-18016-R). Per the protocol, squeeze traps were used to safely restrain the cats to allow for administration of a sedative by participants (IACUC- 18016-R). A combination of morphine (morphine sulfate, Hameln pharma, United Kingdom) (0.3 mg/kg), dexmedetomidine (Dexdomitor, Orion Pharma, Finland) (7 µg/kg), and ketamine (Ketamine hydrochloride, Rotexmedica, Germany) (5 mg/kg) was administered via intramuscular injection through the squeeze trap. Empty traps were weighed and documented and once the cats were trapped, the weight of the combination of the trap and the cat was used to safely determine the weight of the cat.

### Sample collection

Of the 160 cats that were trapped, ten cats were excluded from the study based on the following inclusion/exclusion criteria that reflects the SGU's Small Animal Clinic protocols for spays and neuters

- Age: Presence of permanent teeth are used to estimate the age of cats and they usually erupt by 6 to 7 months of age [62,63]. Based on this information cats with no permanent canines are considered to be younger than six months and were excluded from the study.

- Poor body condition: Patient body condition was assessed using a nine-point body condition scoring (BCS) system, where 1 corresponds to emaciation, 5 corresponds to ideal, and 9 corresponds to morbidly obese [64]. All patients assigned a BCS score of 1 or 2 were excluded from surgery and the study.

- American Society of Anesthesiologists (ASA) Physical Status classification [65]: Patients with ASA status of $\geq 4$ were excluded from the study.

All subjects were trapped as part of the FCP and thus were processed for spaying and neutering. Once sedated, a physical exam was performed by third-year veterinary students supervised by a veterinarian. Age, sex, weight, color, and breed of the cats were determined and recorded. A volume of 1.5 mL blood was collected from the jugular vein into a serum separator tube. After clotting at room temperature for 15 minutes, the serum was transferred to a 1.5 mL microcentrifuge tube. Approximately 3 mL of urine was collected via aseptic cystocentesis or free catch by expressing the urinary bladder. The sample was then transferred to a white top,

additive-free vacuette tube and was centrifuged at 4000 rpm for 5 minutes at room temperature. Approximately 1.5 mL supernatant was aspirated and discarded, and the remaining 1.5 mL was transferred into a white top, additive-free vacuette tube [66]. Both urine and serum samples were stored at -18˚ C until analysis [32].

## DNA extraction

a) Urine samples

Urine sample concentration was performed as described in a study by Zaidi et al. [32]. The 1.5 ml of urine was centrifuged at 20,000xg for 10 minutes. The supernatant was discarded and 800 μL PBS was added to the pellet and vortexed for 15 seconds. The sample was centrifuged again at 20,000xg for 10 minutes, and the supernatant was discarded. Next 140 μL PBS was then added to the pellet and vortexed for 15 seconds, and the resulting solution was used for DNA extraction. DNA extraction was performed as described by Chan et al. [52] using Qiagen QIAamp Viral RNA Mini Kit. The kit is able to inactivate PCR inhibitors present in urine. The urine DNA was extracted according to manufacturer's instructions. Extracted DNA was quantified using the NanoDrop 2000 Spectrophotometer, and then the extracted DNA was stored at -20˚ C until PCR.

b) Serum samples

DNA extraction from serum samples was performed as described by Chan et al. [52] using QIAamp DNA Mini Kit according to manufacturer's instructions. Carrier RNA was used to increase the yield. Extracted DNA was quantified using the NanoDrop 2000 Spectrophotometer and then stored at -20˚ C until PCR.

## PCR and sequencing

Following extraction, DNA was tested by PCR using generic zoonotic *Leptospira*-specific primers that amplify a ~600 bp region of the *rpoB* gene. This gene encodes the beta-subunit of RNA polymerase [67]. The procedure used was according to La Scola et al. [68]. *Leptospira interrogans* DNA obtained from ATCC was used as a positive control. PCR positive amplicons of the expected size were extracted using Qiagen QIAquick Gel Extraction Kit according to manufacturer's directions.

Cleaned amplicons were sent to Molecular Cloning Laboratory in San Francisco, CA, USA for Sanger sequencing using the aforementioned *rpoB* primers. Sequences were analyzed and edited using Chromas (Version 2.6.6) and were compared to known *rpoB* gene sequences in NIH-NCBI GenBank using the Basic Local Alignment Search Tool (BLAST).

## Phylogenetic analysis

After editing the sequences using Chromas, phylogenetic trees were constructed with Grenada cat-derived *Leptospira* sequences and *rpoB* sequences from *Leptospira* species retrieved from GenBank, using MEGA 11. Maximum-likelihood phylogenetic analysis was conducted with the Kimura's 2-parameter nucleotide substitution model, gamma-distributed substitution rates, and an allowance for invariant sites (K2+G+I) and with 1000 bootstrapped replicates. Nodes with bootstrap confidence below 70.0% were condensed in the phylogenetic tree presented. Pairwise sequence alignments were obtained using MEGA 11 software.

### Statistical analysis

Fisher exact test was done to determine statistical significance between serum PCR vs. Urine PCR. A P value of < 0.05 is considered statistically significant while a P value >0.05 is statistically insignificant [69].

## Results

One hundred and sixty animals were trapped at the three sites. Of these, 10 were excluded based on the exclusion criteria. Out of the remaining 150 cats, 53.0% (80/150) were males and 47.0% (70/150) were females (Fig 1A and S1 Table), with the majority aged approximately one year (Fig 1B and S1 Table).

A total of 148 urine samples and 136 serum samples were collected. Fewer serum samples were collected compared to urine samples because some patients had a low BCS or not enough sample for serum extraction. All 148 urine and 136 serum samples underwent DNA extraction followed by PCR and gel electrophoresis.

Overall, *Leptospira* DNA was confirmed in 4/148 (2.7%, 95% CI of 0.1%-5.3% prevalence) in urine and 6/136 (4.4%, 95% CI of 2.7%-6.2% prevalence) in serum samples (Fig 2A, 2B and S2 Table).

Consensus sequences of over 480 bp of the *rpoB* gene were obtained and compared with other *Leptospira* sequences in the database. The *Leptospira* sequences identified from Grenadian cats were 97–100% identical to at least one *Leptospira rpoB* gene sequence in GenBank (Table 1).

The *Leptospira* sequences identified from cats in this study reflect several *Leptospira* species based on a comparison of overlapping regions of the *Leptospira rpoB* gene sequences with known *Leptospira* sequences from bats in Grenada and *Leptospira* sequences in GenBank using MEGA 11 (Fig 3).

All the *Leptospira* isolates from Grenadian cats fall within the same P1 (pathogenic) branch and are closely related to Grenadian bat isolates [70]. In addition, Grenadian cat *Leptospira* isolates form three discrete clusters that are distinct from all previously identified *Leptospira* serovars. These clusters have been designated Clades A, B and C (Fig 3).

Grenada Cat 2 is likely infected with the same or similar strain of *Leptospira* found in Grenada Bat 5. Sequence from Grenada Cat 8 is similar to strain found in Grenada Bat (GBL-GL-

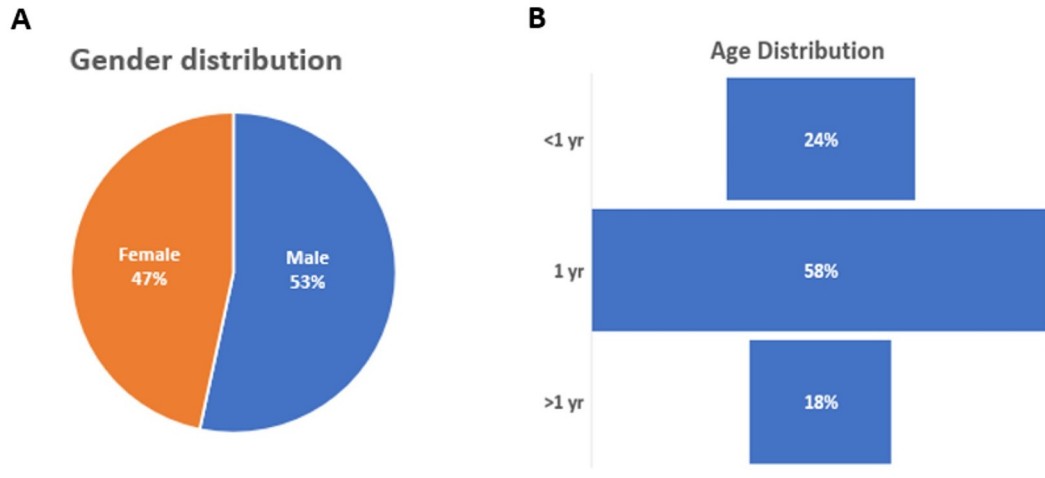

**Fig 1.** (A) Gender distribution, (B) Age distribution.

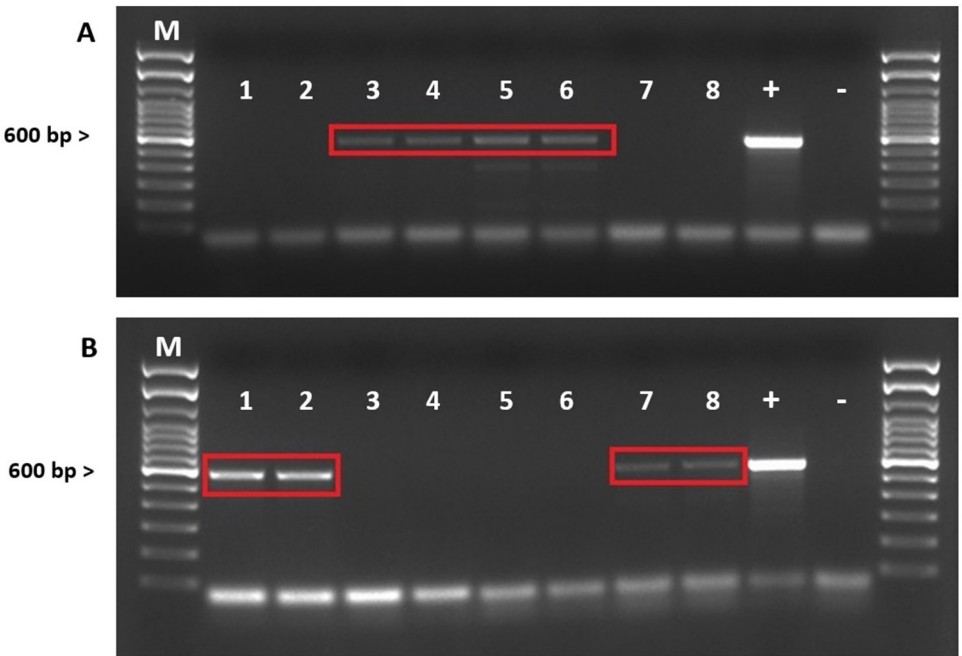

**Fig 2.** *Leptospira* DNA detected in urine (A) and serums (B) samples of cats, with *rpoB* primers.

x33). Grenada Cat 3 is similar to *L. interrogans* serovar Copenhageni. In addition, pairwise comparison of the *Leptospira* DNA sequences derived from cats in Grenada showed close similarity to one another with percentages ranging 88–100% as shown below (Table 1).

## Discussion

The results of this study document the first detection of exposure to and infection with *Leptospira* in the feral Grenadian cat population through PCR. The results show that feral Grenadian cats are infected with *Leptospira* and they are actively shedding pathogenic *Leptospira* genotypes that are phylogenetically related to known pathogenic taxa, some of which may be zoonotic. This has profound implications for public health in Grenada.

DNA extraction from urine and fecal material can be challenging due to rapid degradation and prescence of PCR inhibitors in urine, hence the use of the Qigen Viral kit to help mitigate the degradation these challenges as mentioned in Chan et al [52].

The prevalence of *Leptospira* DNA detected by PCR from urine and serum samples from cats in this study was 2.7% (4/148) and 4.4% (6/136), respectively. The closest match in GenBank for each of the *Leptospira*-positive samples derived from cats in Grenada belonged to P1 (pathogenic *Leptospira*). Based on recommendations of La Scola et al. [68], *Leptospira* genetic sequence comparisons among isolates with *rpoB* gene identity lower than 92.0% represent different *Leptospira* species. Isolates greater than 97.0% identity between partial *rpoB* gene sequences should be regarded as conspecific *Leptospira* [68]. All positive *Leptospira* partial *rpoB* gene sequences derived from cats were 97–100% identical to at least one *Leptospira rpoB* gene sequences in GenBank (Table 1). As observed, the *Leptospira* genotypes described in this study are likely conspecific to known or suspected pathogenic *Leptospira* strains with *rpoB* gene sequences catalogued in GenBank.

Within the Caribbean region, *Leptospira* seroprevalence in cats has been reported as 4.0% in St Kitts [1,11,18], and 12.5% in Trinidad and Tobago [71]. Prior to this study, there were no

**Table 1. Best matches to *Leptospira rpoB* gene sequences from Grenadian cats found in GenBank.**

| Cat *Leptospira* ID number | Species and serovar of closest match in GenBank | Accession number of closest matches | Percent Identity |
|---|---|---|---|
| U85 (30674) | Uncultured *Leptospira* sp. clone GBL-AS-x4 | MG981095.2 | 512/525(98%) |
| | *Leptospira kirschneri* | CP092660.1 | 463/523(89%) |
| | *Leptospira santarosai* | CP028377.1 | 460/523(88%) |
| | *Leptospira interrogans* serovar Bratislava | EU747300.1 | 449/523(86%) |
| U104 (31488) | Uncultured *Leptospira* sp. clone GBL-GL-x14 | MG981100.2 | 547/547(100%) |
| | *Leptospira mayottensis* | CP030147.1 | 484/549(88%) |
| | *Leptospira kmetyi* | CP033614.1 | 482/548(88%) |
| | *Leptospira santarosai* | KJ152440.1 | 482/549(88%) |
| U116 (31061) | *Leptospira interrogans* serovar Copenhageni | CP048830.1 | 521/522(99%) |
| | *Leptospira interrogans* serovar Icterohaemorrhagiae | CP043891.1 | 521/522(99%) |
| | *Leptospira interrogans* serovar Canicola | CP043884.1 | 519/522(99%) |
| | *Leptospira sp.* ADMAS29F | JF718740.1 | 518/522(99%) |
| U122 (30962) | *Leptospira interrogans* serovar Copenhageni | CP048830.1 | 520/522(99%) |
| | *Leptospira interrogans* | CP047508.1 | 520/522(99%) |
| | *Leptospira interrogans* serovar Copenhageni | CP020414.2 | 520/522(99%) |
| | *Leptospira interrogans* serovar Canicola | CP043884.1 | 518/522(99%) |
| S67 (31488) | Uncultured *Leptospira* sp. clone GBL-GL-x33 | MG981111.2 | 522/522(100%) |
| | *Leptospira kmetyi* serovar Malaysia | AB291211.1 | 454/494(92%) |
| | *Leptospira kmetyi* | CP033614.1 | 477/523(91%) |
| | *Leptospira kirschneri* | CP092660.1 | 469/522(90%) |
| S74 (31439) | Uncultured *Leptospira* sp. clone GBL-AS-x4 | MG981095.2 | 530/547(97%) |
| | *Leptospira kmetyi* | CP033614.1 | 488/554(88%) |
| | *Leptospira kirschneri* | CP092660.1 | 487/553(88%) |
| | *Leptospira santarosai* | CP027843.1 | 481/553(87%) |
| S83 (29588) | Uncultured *Leptospira* sp. clone GBL-GL-x33 | MG981111.2 | 529/532(99%) |
| | *Leptospira kmetyi* serovar Malaysia | AB291211.1 | 456/496(92%) |
| | *Leptospira noguchii* | CP091936.1 | 470/532(88%) |
| | *Leptospira interrogans* serovar Hardjo | CP043041.1 | 468/533(88%) |
| S91 (31061) | Uncultured *Leptospira* sp. clone GBL-GL-x33 | MG981111.2 | 547/547(100%) |
| | *Leptospira kmetyi* serovar Malaysia | AB291211.1 | 456/496(92%) |
| | *Leptospira kmetyi* | CP033614.1 | 498/548(91%) |
| | *Leptospira kirschneri* | CP092660.1 | 492/547(90%) |
| S100 (29856) | *Leptospira interrogans* serovar Copenhageni | CP048830.1 | 489/489(100%) |
| | *Leptospira interrogans* serovar Icterohaemorrhagiae | CP043891.1 | 489/489(100%) |
| | *Leptospira interrogans* serovar Canicola | CP043884.1 | 487/489(99% |
| S107 (28987) | Uncultured *Leptospira* sp. clone GBL-GL-x33 | MG981111.2 | 523/525(99%) |
| | *Leptospira kmetyi* | CP033614.1 | 478/526(91%) |
| | *Leptospira interrogans* serovar Hardjo | CP043041.1 | 468/526(89%) |

published reports on *Leptospira* DNA in cats in the Caribbean. In comparison to studies from the rest of the world, the prevalence of *Leptospira* PCR-positives urine samples in this study (4/148; 2.7%) was similar to or within the prevalence reported from Canada (1.6–5.3%) [41], Germany (3.3%) [40] and Malaysia (4.9%) [55]. The prevalence rates of *Leptospira* DNA in urine in the present study were about fourfold lower than prevalence rates reported from shelter cats in Colorado State (11.8%) [20], and much lower than cats in the Taiwan study (67.8%) [52].

The *Leptospira* PCR-positive serum samples in this study (6/136; 4.4%) were similar to rates reported from Worcester County, Massachusetts (3/63; 4.8%) [50] but higher prevalence rates

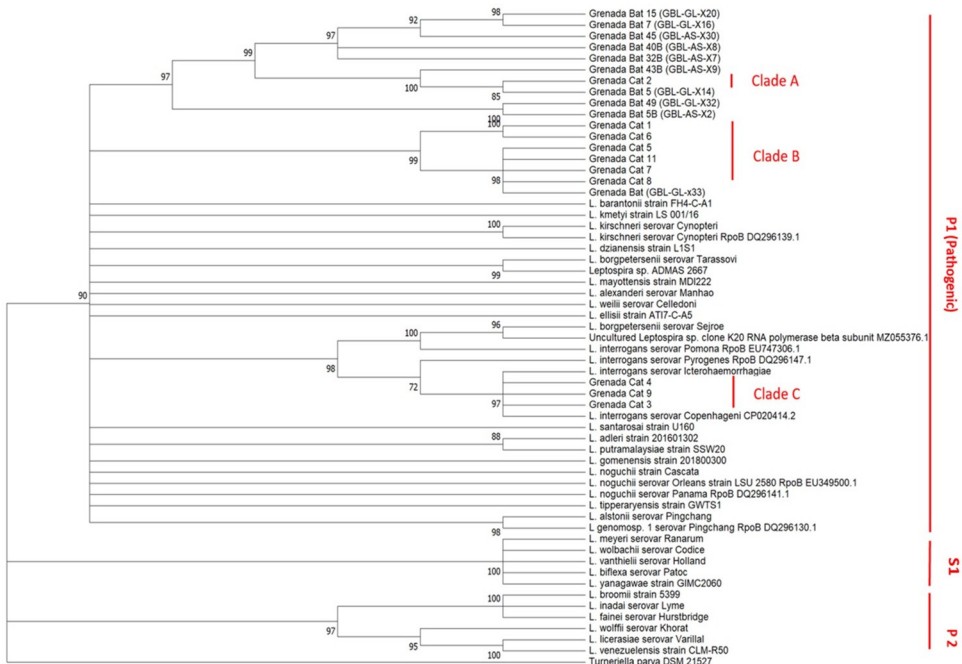

**Fig 3. Phylogenetic tree of Grenadian derived *Leptospira* sequences in relation to sequences in GenBank for known *Lepospira* species.**

were reported from Germany (35/195; 17.9%) [40] and from Quebec, Canada (10/40; 25%) [72]. Further examination of positive *Leptospira* DNA samples showed the following details; four cats were PCR positive via serum only; two cats were PCR positive via urine only and two cats tested positive on both urine and serum for PCR detection of *Leptospira* DNA (S1 Table).

Based on sequencing of DNA from the urine sample, the most likely match on GenBank was *Leptospira interrogans* serovar Canicola, accession no. CP043884.1, with percentage identity of 99.2% (518/522).

There was no difference in the number of male and female cats positive for *Leptospira* on urine/serum PCR, and on average positive cats were about one year old (S1 Table). Several studies have shown prevalence of *Leptospira* infection in young, free roaming feral cats that might be in contact with *Leptospira* reservoir hosts possibly though hunting rodents [20,41,42,48,55,73]. The data from this study is consistent with the demographic described above.

The findings of this study show an overall higher prevalence of *Leptospira* DNA detected via PCR on serum samples than from urine samples, confirming that young feral cats in Grenada have acute infection with circulating organisms in the blood stream. Serum samples with *Leptospira* DNA were 4.4%, in comparison to urine samples with *Leptospira* DNA 2.7%. A possible explanation is the intermittent shedding of *Leptospira* organisms from the kidneys, or low, undetectable amount of *Leptospira* DNA in urine [20,41,54]. Another possible explanation for the low prevalence rate of *Leptospira* DNA in urine could be failure to detect the organism via PCR, due to urine containing compounds and crystals that can inhibit PCR [74]. Furthermore,it is also possible that the serum samples collected were during the acute phase of infection (bacteremia), prior to colonization of the renal tubules [75,76].

The Fisher exact test was used to compare the serum and urine PCR. The results of the study showed p-values >0.05, indicating no significant difference between percent PCR positive serum and urine (S2 Table).

The *Leptospira rpoB* gene sequences derived from Grenadian cats were genetically similar to those obtained from Grenadian bats [70]. This suggests that both cats and bats in Grenada are exposed to the same reservoir host(s) or to the same *Leptospira* contaminated environments such as a water source. It is also possible that the bats in Grenada might be contaminating a water source that the cats are exposed to. Since the Grenadian bats have been documented to carry *Leptospira* [70] is possible that Grenadian cats might be hunting bats, which could lead to infection through ingestion of bats infected with *Leptospira*. Furthermore, Grenada, like many other Caribbean islands, has an abundance of the small Indian mongooses, a reservior host for *Leptospira* spp. [8]. The mongoose could potentially contaminate the environment and lead to exposure for the Grenadian feral cat population. Unfortunately, there are no published ecological studies to lend support to the above scenarios. The identification of *Leptospira* in other animals and humans in Grenada and sequencing would contribute to a better understanding reservoir hosts and transmission senarios.

## Conclusions

Feral cats in Grenada were infected with *Leptospira* and actively shedding the organism through urine. Some cats were infected by known pathogenic serovars (i.e., *L. interrogans*). Some cats were infected with *Leptospira* species that have not been fully described but that are closely related to known or suspected pathogenic and zoonotic *Leptospira*. Sequences of *Leptospira* isolated from cats need to be compared to isolates obtained from human patients to determine if there is a link to human exposure in Grenada.

Although the exact role that cats play in the epidemiology of human leptospirosis is unclear, it is important to recognize the potential risk of exposure to *Leptospira* via feral cats to both human and their pets. Results of this study can help inform the Grenada community about the potential risk of exposure through contaminated water, soil and infected urine. Furthermore, it is important for veterinary practitioners to be aware that cats may present with *Leptospira* infection and to consider leptospirosis as a differential in feline patients. Importantly, the results of this study and other studies confirm pathogenic and zoonotic *Leptospira* in cats. Vaccinating cats against *Leptospira* needs to be considered and potentially become part of core feline vaccines in endemic areas to minimize cat infections and shedding of *Leptospira* with potential human exposure.

## Limitations of the study

One of the limitations of this study reflects the restricted population size and age range, trapping sites and amount of blood sample collected from each cat. All samples collected for this study were obtained through the SGU student organization FCP using established protocols for the trap, neuter, vaccinate and release program. As such the trapping sites and number of cats that could be trapped were limited to areas close to the University in St Georges and thus do not reflect the country as a whole. The amount of blood collected from each cat was limited in order to comply with the IACUC requirements. As a result we did not have enough sample to perform serology as part of this study and in addition to PCR testing. The median age of the cats was estimated to be about 1 year. This might be a limiting factor because the older the feral cats are, the higher the chance for them to have been exposed to the pathogen either by hunting or coming into contact with contaminated water or soil in the environment.

To address these limitations, future studies will include a larger sample size and samples from all parishes in Grenada to get a better representation of the Grenadian feral cat population. An increase in the sample size would likely also expand the age range. To safely obtain more blood from the feral cats for PCR and MAT tests, appropriate aftercare and monitoring

would be required, including hospitalizing the cats, intravenous fluid therapy and monitoring prior to and after the blood draw before release back into the community.

## Supporting information

**S1 Table. Distribution of positive Leptospira in urine and serum samples.**
(XLSX)

**S2 Table. Number of *Leptospira* positive and negative results from cats.**
(XLSX)

## Acknowledgments

We are grateful to all the students and clinicians at the Small Animal Clinic, and Department of Small Animal Medicine and Surgery, SVM, SGU for their assistance in sample collection. The skilled technical assistance provided by Lucinda Ogilvie, Renata Manbodh, Kathrine Moreton and Vanessa Matthew is greatly appreciated.

## Author Contributions

**Conceptualization:** Keith K. Kalasi, Daniel Fitzpatrick, Diana Stone, Talia Guttin, Andy Alhassan.

**Data curation:** Keith K. Kalasi, Daniel Fitzpatrick, Diana Stone, Talia Guttin, Andy Alhassan.

**Formal analysis:** Keith K. Kalasi, Daniel Fitzpatrick, Diana Stone, Talia Guttin, Andy Alhassan.

**Funding acquisition:** Keith K. Kalasi, Andy Alhassan.

**Investigation:** Keith K. Kalasi, Daniel Fitzpatrick, Diana Stone, Andy Alhassan.

**Methodology:** Keith K. Kalasi, Daniel Fitzpatrick, Andy Alhassan.

**Project administration:** Andy Alhassan.

**Supervision:** Daniel Fitzpatrick, Diana Stone, Talia Guttin, Andy Alhassan.

**Validation:** Keith K. Kalasi, Daniel Fitzpatrick, Diana Stone, Talia Guttin, Andy Alhassan.

**Visualization:** Andy Alhassan.

**Writing – original draft:** Keith K. Kalasi.

**Writing – review & editing:** Keith K. Kalasi, Daniel Fitzpatrick, Diana Stone, Andy Alhassan.

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
