## [Decision Letter · Decision Letter 0]

20 Mar 2024

Dear Dr. Alhassan,

Thank you very much for submitting your manuscript "Grenadian cats as potential reservoir for Leptospira" for consideration at PLOS Neglected Tropical Diseases. As with all papers reviewed by the journal, your manuscript was reviewed by members of the editorial board and by several independent reviewers. In light of the reviews (below this email), we would like to invite the resubmission of a significantly-revised version that takes into account the reviewers' comments. 

The authors must improve clarity regarding the significance of their estimated prevalence results in comparison to other studies and areas of the world by contextualizing the techniques used in this study versus the referenced studies that are used for comparison, both in terms of molecular and serological techniques and also in terms of the demographics of the cohort samples used to estimate prevalence. The authors should be clearer about the limited inferences possible from a serological test in this study which has not been adequately validated for cats. The authors should also consider whether the age of the cats trapped in this study may have biased the overall results obtained (i.e., 82% of the 150 cats were 1yo or younger). The authors could comment on whether this trapped cohort of cats can be considered representative of a typical (demographic profile of a) feral cat population in Grenada or other Caribbean islands. Furthermore, the authors should also extend further consideration to whether the small Indian mongoose in Grenada may also have an important role in Leptospira transmission (perhaps in addition to bats), given past and recent reports of association with this host in Grenada and US Virgin Islands (i.e., 10.1371/journal.pntd.0009859 and 10.1016/0035-9203(76)90008-0). 

We cannot make any decision about publication until we have seen the revised manuscript and your response to the reviewers' comments. Your revised manuscript is also likely to be sent to reviewers for further evaluation.

Sincerely,

Yung-Fu Chang

Academic Editor

Amy Gilbert

Section Editor

Reviewer's Responses to Questions

**Key Review Criteria Required for Acceptance?**

**Methods**

-Are the objectives of the study clearly articulated with a clear testable hypothesis stated?

-Is the study design appropriate to address the stated objectives?

-Is the population clearly described and appropriate for the hypothesis being tested?

-Is the sample size sufficient to ensure adequate power to address the hypothesis being tested?

-Were correct statistical analysis used to support conclusions?

-Are there concerns about ethical or regulatory requirements being met?

Reviewer #1: Kalasi et al have compiled a useful set of experiments to assess the prevalence of Leptospira infection in feral Grenadian cats. This manuscript is well organized and clearly written. The introduction, background and methodology are reasonable and very well discussed given the premise of the paper and the importance of the research topic. The figures and tables included in this manuscript are comprehensive, helpful, and well explained. The sample size used in the study (n=150) is sufficient and appropriate to address the hypothesis being tested and the authors have demonstrated a high level of ethical and professional responsibility to follow the inclusion/exclusion criteria for subjects in the study of interest.

Reviewer #2: -Are the objectives of the study clearly articulated with a clear testable hypothesis stated? Yes

-Is the study design appropriate to address the stated objectives? could be improved

-Is the population clearly described and appropriate for the hypothesis being tested? should be improved. the results must be better contextualized.

-Is the sample size sufficient to ensure adequate power to address the hypothesis being tested? 

-Were correct statistical analysis used to support conclusions? yes

-Are there concerns about ethical or regulatory requirements being met? no

**Results**

-Does the analysis presented match the analysis plan?

-Are the results clearly and completely presented?

-Are the figures (Tables, Images) of sufficient quality for clarity?

Reviewer #1: The results of this paper were clearly presented, and the statistical analysis used was correct and unbiased. The figures and tables included in this manuscript are comprehensive, helpful, and well explained.

Reviewer #2: -Does the analysis presented match the analysis plan? yes

-Are the results clearly and completely presented? could be improved

-Are the figures (Tables, Images) of sufficient quality for clarity? could be improved

**Conclusions**

-Are the conclusions supported by the data presented?

-Are the limitations of analysis clearly described?

-Do the authors discuss how these data can be helpful to advance our understanding of the topic under study?

-Is public health relevance addressed?

Reviewer #1: The authors provided a comprehensive conclusion for the problem of interest (prevalence of Leptospira infection in feral Grenadian cats) and for the data presented. They also clearly discussed the limitations of the study and provided very reasonable and acceptable explanations for these limitations. Furthermore, the authors highlighted the novelty of this study in the light of the limited information available for Leptospira , especially zoonotic serovars, exposure and or active infection in feral Grenadian cats and the subsequent public health significance of this problem.

Reviewer #2: -Are the conclusions supported by the data presented? Could be improved

-Are the limitations of analysis clearly described? NO

-Do the authors discuss how these data can be helpful to advance our understanding of the topic under study? Could be improved.

-Is public health relevance addressed? Should be improved.

**Editorial and Data Presentation Modifications?**

Reviewer #1: The following suggested minor wording, grammatical or font modifications are meant to enhance the completeness of this manuscript:

line 32: I recommend changing the word "confirm" to "suggest"

line 112: I recommend starting a new sentence with the word "Furthermore"

line 201: I recommend changing the word "Then" to "Next" to start the sentence

line 213: reference (62) needs to be unbold 

line 299: " La Scola et al. (63) need to be unbold

line 352: I recommend changing the word "Lepto" to "Leptospira" and have it italicized

line 353: the dot (.) after "Furthermore," needs to be removed.

Otherwise very well written manuscript and an important study that highlights the possible significance of Leptospira infection among feral cats in the island of Grenada!

Reviewer #2: (No Response)

**Summary and General Comments**

Reviewer #1: Although the rate/prevalence of Leptospira detection in serum or shedding in urine, through detection of Leptospira genomic contents or measuring of anti-Leptospira antibodies, was relatively low in the subjects involved in this study, the findings clearly suggest that feral Grenadian cats included in this study are either have been exposed or are actively infected with different Leptospira serovars. These findings are novel for the cat populations in Grenada and raise critical questions regarding the role of feral cat population in the spread and transmission of Leptospira infection on the island and the subsequent zoonotic and public health significance of the likelihood and regular interaction between humans and Leptospira-reservoir cats compared to the uncommon interactions between human and several other Leptospira reservoirs such as bats and rodents.

Reviewer #2: Dear Authors and Editors,

Thanks for giving me the opportunity to review this interesting manuscript.

This study describe the possibility that Grenadian cat population may be a reservoir host for zoonotic Leptospira on the island and can be a source of leptospirosis.

General comments: The study aims to characterize the circulation of Leptospira among stray cats in a selected area of Grenada. The animals sampled were very young, which may have resulted in a shorter period of exposure to the pathogen than the average cat's life. Additionally, a serological screening test was used that is not validated for use in cats. When comparing the prevalence of the pathogen in different countries, it is important to consider the diagnostic methods used and other relevant variables. In the following paragraph, I have provided some suggestions and would appreciate a prompt response. If there are any insurmountable obstacles, a 'study limitations' paragraph must be included.

It is also necessary to contextualize the statements by quoting bibliographical references and appropriate citations. Additionally, please ensure correct grammar, punctuation, and proper formatting when indicating the names of the genera and species of microorganisms. Please check all the document for using the correct form (italics) in idicating the genus of the bacteria. Although the writer is fluent in English, revisions are required for form, grammar, and typos.

Overall recomendation: Maior revision required

Abstract

Please check for typos. In addition, you state you had performed the molecular analysis on serum samples.. Is that correct?

Introduction

Line 118- …: I would suggest contextualising the role of the cat as a possible reservoir or as an asymptomatic carrier (difference in epidemiological role). What may be the source of exposure (predatory habits, local wildlife and synanthropy, stray and wild animal population density)?

M&M and Results:

Could selecting young animals (median age 1 y/o) be a limitation of the study? The exposure time to the pathogen may not be representative of the general stray cat population. How can you address this bias?

Line 167: please also state the drugs commercial name, manufacturer and country, as per author guidelines.

Line 197: Which modification?

Line 229: The assessment of serological positivity in cats performed with an antibody immunochromatographic test for dogs could produce inaccurate results. Is it possible that sera could be tested with a more sensitive and specific method (MAT)?

Discussion:

Lines 310-…: Please discuss your results by contextualizing the techniques used by previous studies, both in terms of molecular and serological techniques.

Line 339-340: Disease stage and Leptospira elimination detected by PCR, should be discussed with a view to characterizing a possible carrier or reservoir

Are there any published studies on Leptospira positivity in bats in Grenada? Are there any ecological studies that justify scenario 2? the bibliography needs to be implemented.

It is not clear to me whether the cats were owned (so informed consent from the owner was required) or were strays/feral.

A legend for the size of the population of cats enrolled in the study must be added to Figure 1.

PLOS authors have the option to publish the peer review history of their article (what does this mean?). If published, this will include your full peer review and any attached files.

Reviewer #1: No

Reviewer #2: No
---

## [Decision Letter · Decision Letter 1]

2 Aug 2024

Dear Alhassan,

Thank you very much for submitting your manuscript "Grenadian cats as potential reservoir for Leptospira" for consideration at PLOS Neglected Tropical Diseases. As with all papers reviewed by the journal, your manuscript was reviewed by members of the editorial board and by several independent reviewers. In light of the reviews (below this email), we would like to invite the resubmission of a significantly-revised version that takes into account the reviewers' comments. 

The authors have addressed some of the prior comments, but several critical and additional concerns remain in need of clarity. The authors should be clearer up front with referenced criteria that are used to determine whether a domestic or wild animals is a reservoir for Leptospira. The authors should be clearer about the procedure used to age cats, i.e., what were the units of estimation? The 95% confidence intervals around estimated proportions must be provided throughout the paper for necessary context, especially when making comparisons to other studies. The authors must improve clarity regarding the significance of their estimated prevalence results in comparison to other studies and areas of the world by contextualizing the techniques used in this study versus the referenced studies that are used for comparison, both in terms of molecular and serological techniques and also in terms of the demographics of the cohort samples used to estimate prevalence. It remains unclear why an expected prevalence of 12% among the cats in the population was selected. It is necessary to contextualize the choice and justify the expected population size.

The description of the extraction methods, processing of Leptospira DNA from serum remains inadequate. in addition, it is reported that potentially unsuitable extraction kits were used and the result of the serum PCR is poorly described. The authors are recommended to remove this part or contextualize the use of this approach to a gold standard diagnostic method. 

 We cannot make any decision about publication until we have seen the revised manuscript and your response to the reviewers' comments. Your revised manuscript is also likely to be sent to reviewers for further evaluation.

Sincerely,

Yung-Fu Chang

Academic Editor

Amy Gilbert

Section Editor

Reviewer's Responses to Questions

**Key Review Criteria Required for Acceptance?**

**Methods**

-Are the objectives of the study clearly articulated with a clear testable hypothesis stated?

-Is the study design appropriate to address the stated objectives?

-Is the population clearly described and appropriate for the hypothesis being tested?

-Is the sample size sufficient to ensure adequate power to address the hypothesis being tested?

-Were correct statistical analysis used to support conclusions?

-Are there concerns about ethical or regulatory requirements being met?

Reviewer #1: (No Response)

Reviewer #2: -Are the objectives of the study clearly articulated with a clear testable hypothesis stated? no

-Is the study design appropriate to address the stated objectives? The role of the cats as Leptospira reservoir is poorly contextualised to ecological dynamics and does not relate to a clinical and temporal window of urinary excretion of pathogenic leptospires by stray cats. it is possible that these infected individuals, should they survive infection, have intermittent excretion due to persistence of leptospires in the renal tubules. this hypothesis has not been evaluated or considered.

-Is the population clearly described and appropriate for the hypothesis being tested? The selected animals are very young, reasonably they may have been less exposed to the pathogen. On the other hand, the possibility that they are not perfectly immunocompetent individuals and may be more susceptible to Leptospira infections is a hypothesis that has not been considered

-Is the sample size sufficient to ensure adequate power to address the hypothesis being tested? It is unclear why an expected prevalence of 12% among the cats in the population was selected. It is necessary to contextualise the choice and justify the expected population size

-Were correct statistical analysis used to support conclusions? --

-Are there concerns about ethical or regulatory requirements being met? -- - The study was conducted according to the guidelines approved by the Institutional Animal Care

and Use committee of St. George’s University (IACUC #18016-R dated November 9th 406 , 2018).

DNA extraction and PCR analysis : the description of the extraction methods, processing of Leptospira DNA from serum is inadequate. in addition, it is reported that unsuitable extraction kits (VIRAL) were used and the result of the serum PCR is not even described. I suggest removing this part or contextualising the use of serum by combining two serological methods (ELISA) and a gold standard method (MAT)

**Results**

-Does the analysis presented match the analysis plan?

-Are the results clearly and completely presented?

-Are the figures (Tables, Images) of sufficient quality for clarity?

Reviewer #1: (No Response)

Reviewer #2: -Does the analysis presented match the analysis plan? YES, but I firmly suggest to remove the serum samples molecular analysis. 

-Are the results clearly and completely presented? YES, but I firmly suggest to remove the serum samples molecular analysis. 

-Are the figures (Tables, Images) of sufficient quality for clarity? yes

**Conclusions**

-Are the conclusions supported by the data presented?

-Are the limitations of analysis clearly described?

-Do the authors discuss how these data can be helpful to advance our understanding of the topic under study?

-Is public health relevance addressed?

Reviewer #1: (No Response)

Reviewer #2: -Are the conclusions supported by the data presented? yes

-Are the limitations of analysis clearly described? yes

-Do the authors discuss how these data can be helpful to advance our understanding of the topic under study? yes

-Is public health relevance addressed? yes

**Editorial and Data Presentation Modifications?**

Reviewer #1: (No Response)

Reviewer #2: MAJOR REVISION

**Summary and General Comments**

Reviewer #1: (No Response)

Reviewer #2: Dear Authors and Editors,

 I feel the manuscript could be improved with a few adjustments. I have made some comments in the specific paragraphs which I hope you will find helpful.

It might also be worth checking for typing errors.

 and proper formatting when indicating the names of the genera and species of microorganisms. Please check all the document for using the correct form (italics) in idicating the genus of the bacteria. 

I would be happy to review further versions of the manuscript

PLOS authors have the option to publish the peer review history of their article (what does this mean?). If published, this will include your full peer review and any attached files.

Reviewer #1: No

Reviewer #2: No
---

## [Decision Letter · Decision Letter 2]

11 Nov 2024

Dear Alhassan,

Thank you very much for submitting your manuscript "Grenadian cats as potential reservoir for Leptospira" for consideration at PLOS Neglected Tropical Diseases. As with all papers reviewed by the journal, your manuscript was reviewed by members of the editorial board and by several independent reviewers. The reviewers appreciated the attention to an important topic. Based on the reviews, we are likely to accept this manuscript for publication, providing that you modify the manuscript according to the review recommendations. 

Sincerely,

Yung-Fu Chang

Academic Editor

Amy Gilbert

Section Editor

Reviewer's Responses to Questions

**Key Review Criteria Required for Acceptance?**

**Methods**

-Are the objectives of the study clearly articulated with a clear testable hypothesis stated?

-Is the study design appropriate to address the stated objectives?

-Is the population clearly described and appropriate for the hypothesis being tested?

-Is the sample size sufficient to ensure adequate power to address the hypothesis being tested?

-Were correct statistical analysis used to support conclusions?

-Are there concerns about ethical or regulatory requirements being met?

Reviewer #2: M&M and Results:

Line 240: Please specify which values you considered statistically significant. Please indicate the technic characteristics of the statistic program used.

**Results**

-Does the analysis presented match the analysis plan?

-Are the results clearly and completely presented?

-Are the figures (Tables, Images) of sufficient quality for clarity?

Reviewer #2: M&M and Results:

Line 240: Please specify which values you considered statistically significant. Please indicate the technic characteristics of the statistic program used.

**Conclusions**

-Are the conclusions supported by the data presented?

-Are the limitations of analysis clearly described?

-Do the authors discuss how these data can be helpful to advance our understanding of the topic under study?

-Is public health relevance addressed?

Reviewer #2: Conclusion:

Line 359: It would appear that there is no evidence to suggest that the cat in question poses a direct risk of infection to humans. Indeed, recent literature would seem to indicate that with regard to pathogenic leptospires (see list in various publications), companion animals (dogs, cats in particular) may act as epidemiological sentinels, given that the disease is considered to be an environmental disease. This sentence must be revised (“It is important to educate the public about the potential risk of exposure to Leptospira via feral cats.”)

**Editorial and Data Presentation Modifications?**

Reviewer #2: (No Response)

**Summary and General Comments**

Reviewer #2: Dear Authors and Editors,

I am grateful for the chance to revise the manuscript once more.

The manuscript has been revised and improved. A few comments and suggestions for further enhancements remain.

Overall recomendation: Minor revision required

Introduction

Line 110: Please add more updated bibliography (2, 18, 20, 21, 22, ..)

Line 118- “Hence cats serve as a source of infection and a environmental sentinel for Leptospira (20).”

Line 128: typing errors

M&M and Results:

Line 240: Please specify which values you considered statistically significant. Please indicate the technic characteristics of the statistic program used.

Conclusion:

Line 359: It would appear that there is no evidence to suggest that the cat in question poses a direct risk of infection to humans. Indeed, recent literature would seem to indicate that with regard to pathogenic leptospires (see list in various publications), companion animals (dogs, cats in particular) may act as epidemiological sentinels, given that the disease is considered to be an environmental disease. This sentence must be revised (“It is important to educate the public about the potential risk of exposure to Leptospira via feral cats.”)

PLOS authors have the option to publish the peer review history of their article (what does this mean?). If published, this will include your full peer review and any attached files.

Reviewer #2: No

Figure Files:

Data Requirements:

Reproducibility:

References

---

## [Editor Report · Decision Letter 3]

16 Dec 2024

Dear Alhassan,

We are pleased to inform you that your manuscript 'Grenadian cats as potential reservoir for Leptospira' has been provisionally accepted for publication in PLOS Neglected Tropical Diseases.

Best regards,

Yung-Fu Chang

Academic Editor

Amy Gilbert

Section Editor

Shaden Kamhawi

co-Editor-in-Chief

Paul Brindley

co-Editor-in-Chief

---

## [Editor Report · Acceptance letter]

20 Dec 2024

Dear Associate Professor Alhassan,

We are delighted to inform you that your manuscript, "Grenadian cats as potential reservoir for Leptospira," has been formally accepted for publication in PLOS Neglected Tropical Diseases.

Best regards,

Shaden Kamhawi

co-Editor-in-Chief

Paul Brindley

co-Editor-in-Chief
